# Anti-Cancer Activity of Porphyran and Carrageenan from Red Seaweeds

**DOI:** 10.3390/molecules24234286

**Published:** 2019-11-25

**Authors:** Zhiwei Liu, Tianheng Gao, Ying Yang, Fanxin Meng, Fengping Zhan, Qichen Jiang, Xian Sun

**Affiliations:** 1School of Pharmacy and Food Sciences, Zhuhai College of Jilin University, Zhuhai 519041, China; zwliumost@126.com (Z.L.); mfx@jluzh.edu.cn (F.M.); ZhanFengping@163.com (F.Z.); 2School of Environment and Energy, South China University of Technology, Guangzhou 510006, China; 3Institute of Marine Biology, College of Oceanography, Hohai University, Nanjing 210017, China; gaotianheng928@hhu.edu.cn; 4Freshwater Fisheries Research Institute of Jiangsu Province, 79 Chating East Street, Nanjing 210017, China; sanbao1120@gmail.com; 5School of Marine Sciences, Sun Yat-Sen University, Guangzhou 510275, China

**Keywords:** seaweed, porphyran, carrageenan, anti-cancer

## Abstract

Seaweeds are some of the largest producers of biomass in the marine environment and are rich in bioactive compounds that are often used for human and animal health. Porphyran and carrageenan are natural compounds derived from red seaweeds. The former is a characteristic polysaccharide of *Porphyra*, while the latter is well known from *Chondrus*, *Gigartina*, and various *Eucheuma* species, all in Rhodophyceae. The two polysaccharides have been found to have anti-cancer activity by improving immunity and targeting key apoptotic molecules and therefore deemed as potential chemotherapeutic or chemopreventive agents. This review attempts to review the current study of anti-cancer activity and the possible mechanisms of porphyran and carrageenan derived from red seaweeds to various cancers, and their cooperative actions with other anti-cancer chemotherapeutic agents is also discussed.

## 1. Introduction

Cancers are serious diseases of various etiologies, especially that of unhealthy eating habits and lifestyle. In 2018, about 9.6 million cancer-related deaths and 18 million new cases were estimated by the World Health Organization (WHO) [1]. Uncontrolled growth, invasiveness, and metastasis are characteristics of tumor cells evoked by acquired genetic changes [2]. With tumor development, unbalanced programmed cell death, disordered signaling pathways, angiogenesis, and poor immune response disrupt various homeostatic pathways. Such deregulated pathways are the main targets of cancer treatment by chemotherapy [3]. According to the characteristics and stage of the tumor, combined therapy is applied in cancer treatment including surgery, chemotherapy, radiation therapy, and immunotherapy. The ultimate aim of all treatments is to destroy the tumor cells in the achievement of cancer treatment, while avoid damaging normal cells as far as possible. Unfortunately, severe side effects are often unavoidable, limiting the efficacy of treatment. Chemotherapy is commonly and effectively used in cancer therapeutics, exerting cytotoxicity on rapidly dividing and proliferating cells, not only including malignant cells, but also normal cells with high-proliferating potential. Thus, chemotherapy usually brings serious side effects including anemia, appetite loss, delirium, alopecia, peripheral neuropathy, and irreversible damage to vital organs [4]. In addition, drug tolerance is also an issue in cancer treatment, which would weaken the treatment effects. Targeted therapy could avoid the side effects in part, but not always completely. Monoclonal antibodies are generally safer than chemotherapy only with mild allergic reactions such as urticaria for the design of a specifically targeted treatment to the cancer antigens located on tumor cells. However, severe reactions are still hard to avoid. For instance, patients who have a high burden of tumor cells in their circulation would face a high risk of tumor lysis syndrome and other severe complications such as anaphylactic reactions and myocardial infarction in occasional cases [5]. Therefore, developing low side-effect and better-tolerated anti-cancer agents is compelling.

Natural products are attractive sources for the development of new medicinal and therapeutic agents for their cell selective and fewer adverse effects. In this context, it is significant to develop natural products in cancer treatment. According to reports, natural origins are the main origins for approved drugs in the treatment of cancer, occupying almost 60% [6]. Though the development of marine natural products is still in its embryonic stage, it is anticipated that marine natural products will become an invaluable source for the development of new medicinal and therapeutic agents in cancer treatment because of their large habitat (covering ~70% of the Earth’s surface), high biodiversity (95% of world biodiversity), and the specific conditions under which some species live [7,8]. It has great scope in which discover new anti-cancer medicine for large production, biological activity, and have unique chemicals. Over the last few decades, pharmaceutical companies and academic institutions have made significant efforts in deriving and identifying new marine products from marine organisms, with more than 3000 new anti-cancer compounds [9]. Of particular interest are the products derived from seaweeds with anti-cancer potential in natural marine products.

Seaweeds are widely distributed in cold, temperate, and tropical zones and play vital roles in sustaining the biodiversity and ecology of marine ecosystems. Several species of economic value such as *Laminaria*, *Porphyra*, and *Gracilaria* are cultured in the coastal waters of many countries [10]. Seaweeds are low in lipids, rich in proteins, minerals, vitamins, antioxidants, phytochemicals, polyunsaturated fatty acids, and are also a source of a vast number of novel compounds with unique health benefits such as essential amino acids and their proteins as well as essential minerals [11,12]. Epidemiological studies have shown that a seaweed-rich diet reduces the incidence of obesity, cancer, and heart and cerebrovascular diseases [13]. A large number of studies have uncovered the anti-cancer activities of seaweeds and numerous seaweed-derived compounds that have been shown to be effective through multiple mechanisms such as the inhibition of cancer cell growth, invasiveness and metastasis as well as by the induction of apoptosis in cancer cells. Some of the substances have been developed into drugs for cancer treatment [3,14,15,16,17]. In recent years, natural compounds extracted from marine algae have been proposed as effective in inhibiting tumor growth, adhesion, invasion, and migration [15].

Polyphenols and sulfated polysaccharides are the predominant belongings of seaweed, possessing an array of pharmacological properties [6]. Polysaccharides are found in the intracellular space and in the fibrillar cell walls of seaweeds [2]. Recently, considerable attention has been focused on polysaccharides isolated from natural sources. Such polysaccharides, which are the main storage compounds in seaweed, are polymers of hexoses or other monosaccharides with antioxidant, anti-cancer, anti-coagulant, and anti-inflammatory properties and are widely included in commercial products [18,19,20]. Small differences in structures in these polysaccharides determine their distinctive properties. These large molecules are divided into either homopolysaccharides or homoglycans and heteropolysaccharides or heteroglycans. Both are distinguished by a monomeric unit, which is of only one kind in the former such as cellulose and starch, or two or more kinds in the latter. Additionally, the polymers are divided into brown, red, green, and blue polysaccharides, according to the type of seaweed from which they are derived. The former two polysaccharides have attracted more attention and are widely applied. Alginic acid, fucoidan (sulfated fucose), and laminaran (β-1,3 glucan) are derived from brown seaweed. Agars, carrageenans, xylans, floridean starch (amylopectin-like glucan), water-soluble sulfated galactans, and porphyrans are from red algae. Green seaweeds contain sulfuric acid polysaccharides, sulfated galactans, and xylans. Seaweed polysaccharides are diverse and characteristic of specific species and vary with season. Up to 76% of the dry weight is polysaccharide in some genera such as *Ascophyllum*, *Porphyra*, and *Palmaria* [21]. This review attempts to review the current study of anti-cancer activity and the possible mechanism of porphyran and carrageenan derived from red seaweeds to various cancers, and their cooperative action with other anti-cancer chemotherapeutic agents is also discussed. The keywords, “red seaweed”, “cancer”, “polysaccharide”, “porphyran”, and “carrageenan” were searched in “Google Scholar” and “Web of Science” in the period between 1980 and 2019.

## 2. Anti-Cancer Activity from Red Seaweeds

Edible red seaweeds have been considered as a healthy and beneficial food in Asia such as Japan, China, Thailand, and South Korea for a long time. Red seaweed cultivation has significantly grown rapidly since the early 20th century due to the continuous increase in demand for food and industry [10]. *Kappaphycus*, *Eucheuma*, *Gracialria,* and *Porphyra* are the main species largely cultivated in Indonesia and China. Bioactive compounds of seaweeds are synthesized in accordance with seaweed growth stage and the ability to interact with environmental changes such as radiation, water pressure, and salinity [7]. Phycobiliproteins, carotenoids, pigments, terpenes, polyphenols, phlorotannins, and polysaccharides are the major contributors to seaweeds, with various types and amounts in different species [3,11,22]. Terpenes, polysaccharides, and polyphenols are of major interest for their anti-cancer activity [2,3,23].

The anti-cancer effects of seaweed could be as nutrients and cytotoxic properties [19]. As a nutrient source, seaweed limits the development of cancers, probably by enhancing antioxidant properties. Through the mechanisms of carcinogenesis promoted by oxidative processes, it is obvious that antioxidants play vital roles in the later stages of cancer development. Thus, antioxidants are deemed as a feasible manner to regress premalignant lesions and inhibit cancer development [6]. Meanwhile, natural seaweed products have cytotoxic properties when concentrated. Researchers have reported that a sulfated polysaccharide from *Champia feldmannii* did not show obvious in vitro cytotoxicity, but was antitumor against sarcoma 180 in mice, probably associated with its immune stimulating properties [24]. A sulfated polysaccharide isolated from *Gracilaria lemaneiformis* exhibited remarkable anti-cancer and immunomodulatory activities against transplanted H22 hepatoma cells in ICR (Institute of Cancer Research) mice. Marked inhibition of tumor growth, promotion of splenocyte proliferation, macrophage phagocytosis, and the level of increments of IL-2 and CD8^+^ T cells in blood [25] were all affected. The in vitro and in vivo anti-cancer studies of the sulfated polysaccharide isolated from *C. feldmannii* was carried out in Swiss mice. Though the in vitro cytotoxicity of the polysaccharide was not significant, the in vivo anti-cancer effect was measurable. The increased immune stimulation including increasing both the production of specific antibodies and the production of OVA-specific antibodies as well as inducing a discreet hyperplasia of lymphoid follicles of the white pulp in the spleen, were associated with anti-cancer activity [24]. Anti-cancer effects were also demonstrated in the polysaccharides derived from other seaweeds, especially fucoidan from brown seaweeds. The anti-cancer activity of fucoidans has been reported in many types of cancers such as lung cancer [26,27], gastric cancer [26], breast cancer [28], and liver cancer HepG2 cell [29]. In the following section, porphyran and carrageenan, the polysaccharides derived from red seaweeds, are described in detail.

Anti-cancer activity has also been proven in other compounds. Terpenes and their derivations, halogenated monoterpenes, are compounds of seaweeds, usually as secretin outside the cell to defend against environmental stress with high anti-cancer activity. The halogenated monoterpene halomon [6(*R*)-bromo-3(*S*)-bromomethyl)-7-methyl-2,3,7-trichloro-1-octene] was the first monoterpene isolated from *Portieria hornemannii* [30] with sub-micromolar activity (IC_50_ ≤ 0.9 μM) against at least one cancer cell line including renal-, brain-, and colon-derived solid tumor cell lines [31]. Other halogenated monoterpenes isolated from the red seaweeds *Plocamium suhrii* and *Plocamium cornutum,* showed greater antiproliferative activity on an esophageal cancer cell line (WHCO1) when compared with cisplatin, well-known as an anti-cancer drug [32]. Phenolic compounds are composed of a single aromatic ring and possess large, broad biological activities due to the ring bearing one or more hydroxyl groups [4,33]. Bromophenols, polyphenolics compounds with one or more bromine substituents, are most commonly found in red seaweeds [34]. A polyphenol-rich extract from *Eucheuma cottonii* was proven to have selective cytotoxicity in estrogen-dependent MCF-7 and estrogen-independent MB-MDA-231 human breast-cancer cells (IC_50_ values of 20 and 42 μg/mL, respectively) depending on dose. Polyphenol showed anti-cancer activity by inducing apoptosis, downregulating the endogenous estrogen biosynthesis, and improving antioxidative status [35]. Additionally, polyphenols from red seaweed *Corallina officinalis* have been applied in nano-biotechnology and biosynthesized to gold nanoparticles as a reducing and stabilizing agent. The gold nanoparticles showed cytotoxic activity against MCF-7 cells depending on the dose of gold nanoparticles and the polyphenol content [36]. Pheophorbide a (Pa) is a product of chlorophyll breakdown, having been applied in the photodynamic therapy of cancer as a chlorine-based photosensitizer [37]. Pa-mediated photodynamic therapy (PDT) was used in treating 7,12-dimethylbenz[*a*]anthracene (DMBA)/12-*O*-tetradecanoylphorobol-13-acetate (TPA)-induced mouse papillomas with marked downregulation of proliferating cell nuclear antigen expression [37]. The Pa isolated from *Grateloupia elliptica* was proven to have specific anti-cancer activity toward various cancer cells lines including B16-BL6, HeLa, SiHa, SK-OV-3, and U87MG cells, especially in U87 MG glioblastoma cells [6]. The Pa induced G0/G1 arrest of U87 MG cells in the absence of direct photo-irradiation, causing late apoptosis and DNA degradation under dark conditions. These results suggest that Pa isolated from *G. elliptica* is a potential glioblastoma-specific anti-cancer agent without side effects on normal cells.

## 3. Porphyran

Porphyran is a characteristic polysaccharide of *Porphyra*, also a red seaweed. Various species are ‘‘Nori’’, which is marketed in sheets of dried seaweed and is popular in East and Southeast Asia as well as globally, especially as a wrap for sushi. Porphyran is a galactose, highly substituted by the 6-*O*-sulfation of L-galactose units and 6-*O*-methylation of d-galactose units (Figure 1) [38,39]. Various methods including hot water extraction, radical degradation, and ultrasonic treatment have been used to extract porphyrans from red seaweeds. Porphyrans have been reported to be hypolipidemic, anti-cancer, and anti-inflammatory in human beings. Porphyran inhibits NO production in macrophages by blocking NF-B activation in the mouse macrophages of RAW264.7 cells that were stimulated with lipopolysaccharides. This may explain some of the anti-inflammatory effects of porphyran [40]. It has been reported that porphyrans have the potential to prevent hyperlipidemia due to its excellent antioxidant activities in mice [14]. Previous studies have shown that porphyrans inhibited lipid synthesis in HepG2 cells and also decreased apolipoprotein B100 secretion, realizing its hypolipidemic effect [41]. Oral porphyran alleviates liver damage induced by the high-fat diet of ICR mice, implicating the use of porphyran as a dietary hypolipidemic component [42]. Furthermore, porphyran was proven to be effective and potential in anti-cancer by various studies (Table 1, Figure 2).

Generally, porphyran is non-toxic on normal cells, although toxic for cancer cells, and induces cell death in a dose-dependent manner [43]. In vitro anti-proliferative activity of crude and purified porphyran, also in a dose-dependent manner, was reported in HT-29 colon cancer cells and AGS gastric cancer cells. The polysaccharide portion of the crude porphyran was thought to account for anti-proliferative activity via apoptosis, as indicated by increased caspase-3 activity [44]. The anti-cancer activity of porphyran against Ehrlich carcinoma and Meth-A fibrosarcoma has been demonstrated in mice tumor models [45,46]. Similar results have been reported in cancer cells of AGS and HT-29, the proliferation of which was arrested by *Porphyran*-*chungkookjang*, prepared by adding 5% *(w/w)* porphyran into fermented *Bacillus subtilis* [47]. The methanol extract of *porphryan-chungkookjang* showed higher anti-cancer effects than the *chungkookjang*.

One study revealed that AGS gastric cancer cells were effectively controlled by porphyrin, which decreased cell proliferation and induced apoptosis. Negative regulation of IGF-IR phosphorylation and activation of caspase-3 is a porphyran effect [47]. Other investigations showed that a polysaccharide from *Porphyra yezoensis* arrested the cancer cell cycle at either the G0/G1 or G2/M check points [48]. Cell proliferation was also inhibited in the HeLa line, which were induced by porphyran. The cell cycle was blocked in the G2/M phase by regulating and controlling the expression of p21, p53, cyclin B1, and CDK1 [49].

There is growing evidence that the biological activities of polysaccharides are dependent on their molecular weight, conformational state, chemical components, and glycosidic bonds [50,51]. Molecular weight is especially important because it is related to viscosity, water-solubility, conformation, and other basic properties of polysaccharides [38,52]. Lower molecular weight porphyrans have a higher antioxidant activity [39,53]. Although discolored due to a lack of nutrients that reduces their commercial value considerably, in cultured *P. yezoensis*, a higher level of porphyran was found in the discolored organisms. It has greater ROS-scavenging activity, likely due to the lower mean molecular mass of the porphyran [54]. Additionally, oligo-porphyran, the acid hydrolysis product of porphyran, has the potential to prevent and treat various pathologies such as Parkinson’s disease and acute renal failure. Previous studies have suggested that oligo-porphyran protects renal morphology and function in rats with renal impairment [39]. They also ameliorate neurobehavioral defects by regulating the PI3K/Akt/Bcl-2 pathway in Parkinsonian mice [55]. The anti-cancer response to porphyran shows varying results. For example, porphyran derived from *P. yezoensis* was degraded by gamma irradiation so that the exposure dose of irradiation was higher and the molecular weight of porphyrans lower [49]. No significant changes in the contents of sulfate, monosaccharide composition, and 3,6-anhydroanhydro-α-L-galactose were detected in the three polysaccharides. These inhibited the cancer cell lines of HeLa and Hep3B more effectively than the degraded products. This discovery contradicts other studies that concluded that lower molecular weight porphyrans exert more anti-cancer activity [38]. The relationship between the molecular weight of porphyrans and their anti-cancer activity along with their conformation should be studied further.

## 4. Carrageenan

Carrageenan is a highly sulfated polysaccharide found in *Chondrus*, *Gigartina*, and various *Eucheuma* species in the red algal family Rhodophyceae [38]. It is widely used in food and pharmaceutical industries as a stabilizer, a gelling agent, thickener, binder, and additive [56]. D-galactopyranosyl with one or two sulfate groups is the base unit of carrageenans, linked via alternated (1→3)-β-d-and (1→4)-α-d-glucoside (Figure 3) [56,57,58]. The number and position of the sulfate groups divide carrageenans into α-carrageenan, β-carrageenan, γ-carrageenan, δ-carrageenan, θ-carrageenan, ι-carrageenan, κ-carrageenan, λ-carrageenan, μ-carrageenan, and ν-carrageenan (Figure 3), and of these, κ-, ι-, and λ-carrageenans, are of commercial significance [59]. Acidic hydrolysis is effective in analyzing their structures through reductive hydrolysis [60,61], and enzymatic hydrolysis is preferred in industrial production [62]. Although carrageenan is generally regarded as safe [38], its consumption is reported to cause colitis [63,64,65]. It is also reported to induce paw edema and pleurisy in experimental rats, which is widely used to study anti-inflammatory activity and the mechanisms involved in inflammation [66,67,68,69,70]. Carrageenan induces thrombosis in a tail thrombosis model and is frequently used to study the mechanisms of antithrombosis and thrombolysis in small laboratory animals [71,72,73]. Growing evidence suggests the anti-cancer ability of carrageenan (Table 2, Figure 2).

Natural anti-cancer defense mechanisms in the host play an important role in cancer treatment combined with a variety of therapeutic approaches including new anti-cancer drugs that enhance immunity [74]. Seaweed polysaccharides are reported to regulate immune responses by activating immune cells and other generalized immune responses. Immunomodulating activity induced by carrageenan has been studied in the treatment of tumors by several researchers. λ-carrageenan was reported to inhibit tumor growth in B16-F10- and 4T1-bearing mice through intratumoral injection [75]. Meanwhile, immune response to the tumor was enhanced by promoting tumor-infiltrating M1 macrophages in the spleen, which secreted higher levels of IL17A in the spleen and TNF-α in the tumor. Humoral and cell-mediated immunity in S180-bearing mice was also reported to be enhanced by carrageenan oligosaccharides extracted from *Kappaphycus striatum* and led to potent tumor therapeutic activity [76].

The selective cytotoxic effects of carrageenans on cancer cells have been demonstrated in several investigations. Such studies have shown that concentrations of 250–2500 μg/mL of both κ-carrageenan and λ-carrageenan inhibited human cervical carcinoma cells by not only arresting the cell cycle at specific phases, but also by delaying the time of it [56]. κ-carrageenan delayed the cell cycle in the G2/M phase while λ-carrageenan delayed both G1 and G2/M phases. However, κ-selenocarrageenan (i.e., κ-carrageenan with selenium) is anti-proliferative on the human hepatoma cell line. It blocks the cell cycle in the S phase [77]. However, native ι-carrageenan showed no significant anti-proliferation in the human osteosarcoma cell line in either in vitro or in vivo assays. Degraded ι-carrageenan [78] suppressed tumor growth, induced apoptosis, and arrested the G1 phase, which improved the survival rate of tumor-bearing mice. Downregulation of the Wnt/β-catenin signaling pathway was responsible for that.

Angiogenesis plays a vital role in cancer development. Therefore, anti-angiogenic activity is widely explored in cancer treatment. As they have better anti-angiogenic activity than the standard compound, suramin, carrageenans have been defined as angiogenesis inhibitors [70,80,81]. The anti-angiogenic activity of κ-carrageenan oligosaccharides was shown in ECV304 cells and the CAM (Chicken chorioallantoic membrane) model to inhibit the proliferation, migration, and tube formation of cells [79]. Moreover, the oligosaccharides inhibited new blood vessel formation with the negative regulation of human VEGF, bFGF, bFGFR, and CD105 in MCF-7 xenograft tumors. The negative effect on tumor blood vessel endothelial cell differentiation was also demonstrated in human umbilical vein endothelial cells and were affected by λ-carrageenan oligosaccharides at relatively low concentrations (150–300 μg/mL) [82] by the downregulation of intracellular matrix metalloproteinase (MMP-2) expression.

The biological activities of sulfated polysaccharides are a function of structural features such as the amount and position of sulfation and molecular weight. That is, the chemical modification of carbohydrates leads to variations in their biological activities [83]. For example, λ-carrageenan can be degraded into five products, all with different molecular weights and all showing anti-cancer effects, probably through immunomodulation. Lower molecular weight products, 15 and 9.3 kDa, showed higher anti-cancer and immunomodulation effects [83]. Selective chemical sulfation in the carrageenan backbone plays a measurable effect on its anticoagulant activity, which would be promoted by the substitution by sulfate at C6 of β-d-Galp and C2 of 3,6-anhydro-α-d-Galp units [84]. Another example, sulfate at C2 of the β-d-GalAp units, showed a more positive effect on the anticoagulation than at C4. Additionally, the partially oxidized molecule promoted the anticoagulant effect of the κ-carrageenan derivative more than the fully oxidized molecule [85]. Anti-cancer and immunomodulation activities of κ-carrageenan oligosaccharides from *Kappaphycus striatum* were enhanced by sulfation, acetylation, and phosphorylation where the sulfated derivative was the most effective. Chemical modifications also promoted oxidant activity by κ-carrageenan oligosaccharides [86].

## 5. Combination with Conventional Anti-Cancer Drugs

Toxicity analyses have proven that polysaccharides are potent anti-cancer agents and effective adjuvants in cancer immunotherapy. 5-Fluorouracil (5-Fu), a thymidylate synthase inhibitor, has been widely used to treat cancer for several decades. However, it is limited by undesirable side effects [87,88,89]. When the drug was fixed at the 6-position with low molecular weight porphyran in order to obtain a water-soluble macromolecule prodrug, it led to a slow release of 5-Fu and prolonged the duration of anti-cancer activity and reduced the side effects [88]. The mixture and conjugate enhanced the anti-cancer activity of 5-Fu and immunocompetence recovered the damage in transplanted S180 tumor mice. The medical effect of the λ-carrageenan on anti-cancer activity and immunosuppression by 5-Fu were explored on transplanted S180 tumor mice [90]. Though the individual use of the λ-carrageenan sample or 5-Fu at low dose only exerted low anti-cancer activity, a mixture of the two samples at the same dose increased the activity. Meanwhile, λ-carrageenan enhanced immunocompetence that had been damaged by 5-Fu by increasing the weight of the spleen, activating lymphocyte proliferation, recovering the level of TNF-α, and reactivating the decreased spleens and white pulps. Similar research supports this result in H-22 tumor mice [91].

Gold nanoparticles (AuNPs) have been widely used in catalysis, photothermal therapy, and targeted drug delivery [92]. The κ-carrageenan oligosaccharide was reported as a reducing and capping agent to prepare AuNPs, which showed significant cytotoxic activities to HCT-116 and MDA-MB-231 cells [93]. Furthermore, maghemite nanoparticles have been reported to be electrostatically entrapped by ι-carrageenans in the sulfate groups [94]. In vitro anti-cancer efficacy of the biocompatible ι-carrageenan-γ-maghemite nanocomposite was demonstrated in the human colon cancer cell line by inducing cell apoptosis by following the ROS-mediated mitochondrial pathway, combined with downregulation of the expression levels of mRNA of XIAP and PARP-1, and the upregulation of caspase3, Bcl-2, and Bcl-xL.

## 6. Conclusions

The ideal cancer treatment eradicates tumor cells without damage to healthy tissues. Due to the side effects of current treatments, more attention is being paid to the selective toxicity of seaweed polysaccharides that are nontoxic to normal cells, but toxic to tumor cells. Several in vitro and in vivo studies have demonstrated that porphyrans and carrageenans have strong anti-cancer properties. Moreover, when combined with conventional drugs, these polysaccharides not only showed more effective anti-cancer activity, but also enhanced immunocompetence that had been damaged by drugs such as by increasing the weight of the spleen, activating lymphocyte proliferation, recovering the level of TNF-α, and reactivating the decreased spleens and white pulps.

## Figures and Tables

**Figure 1 molecules-24-04286-f001:**
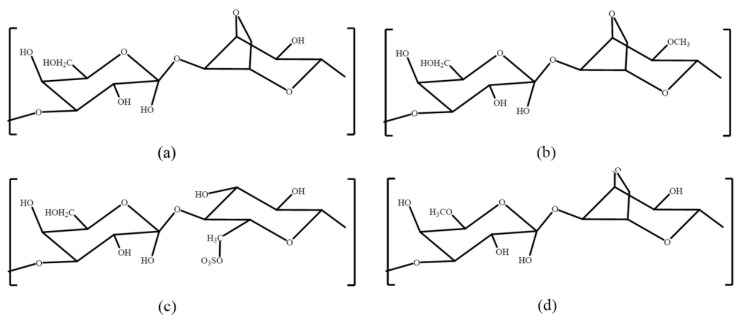
Typical repetitive structures in porphyran [38]: (**a**) G-A; (**b**) G-A2M; (**c**) G-L6S; (**d**) G6M-A. G: 1,3-linked β-d-galactose; A: 1,4-linked 3,6-anhydro-α-l-galactose; A2M: 1,4-linked 2-*O*-methyl-3,6-anhydro-α-l-galactose; L6S: 1,4-linked α-l-galactose 6-sulfate; G6M: 1,3-linked 6-*O*-methyl-β-d-galactose.

**Figure 2 molecules-24-04286-f002:**
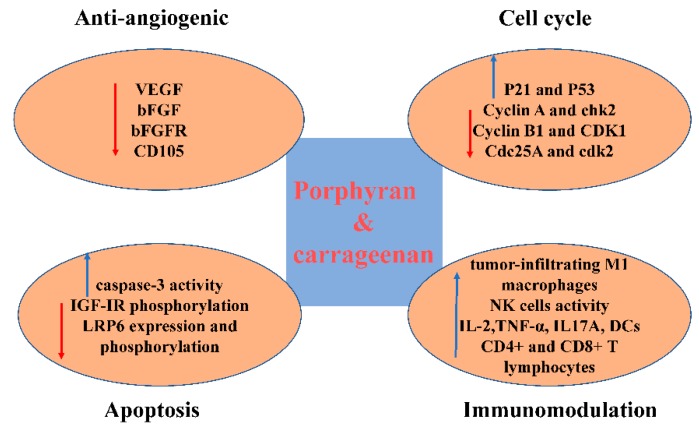
Possible mechanism in the anti-cancer activity of porphyran and carrageenan.

**Figure 3 molecules-24-04286-f003:**
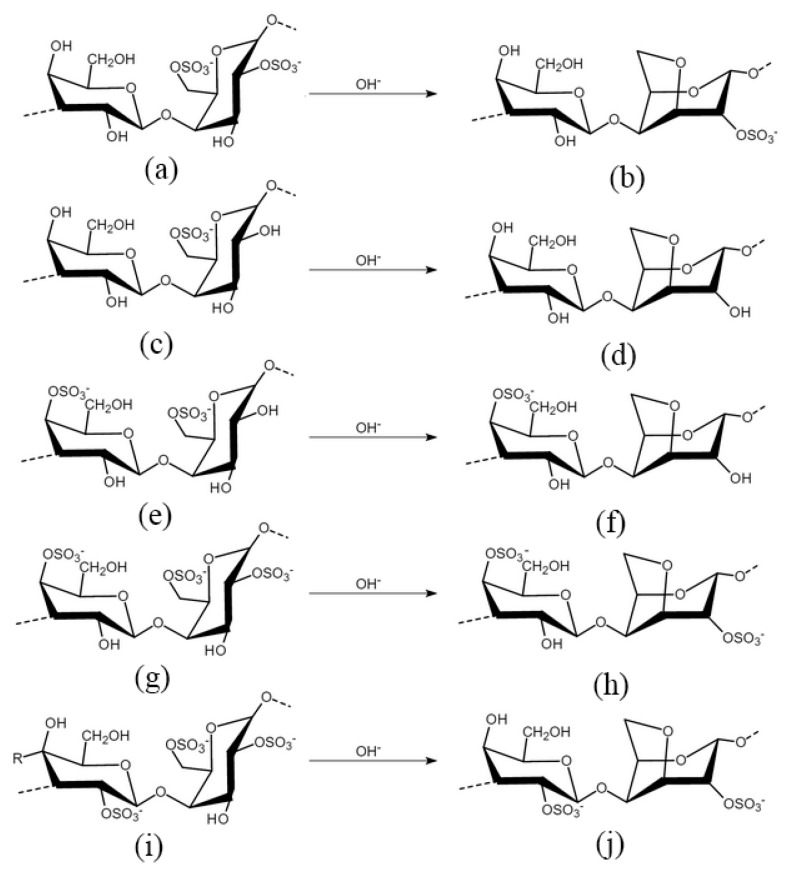
Chemical structure of carrageenans [64]: (**a**) δ-carrageenan; (**b**) α-carrageenan; (**c**) γ-carrageenan; (**d**) β-carrageenan; (**e**) μ-carrageenan; (**f**) κ-carrageenan; (**g**) ν-carrageenan; (**h**) ι-carrageenan; (**i**) λ-carrageenan; and (**j**) θ-carrageenan.

**Table 1 molecules-24-04286-t001:** Anti-cancer activity and possible mechanisms of porphyran.

Source	Target	Type of Activity	Possible Mechanisms	References
*P. yezoensis*	Mice implanted with Ehrlich carcinoma and Meth A fibrosarcoma	Appreciable inhibition of tumor growth	Not referred	[45,46]
	AGS and HT-29 cancer cells	Antiproliferation		[47]
	SGC-7901 and 95D cancer cell lines			[38]
	Hep3B cells	Antiproliferation and cell cycle blocked in the G2/M phase	Upregulation of p21 and p53, while negatively regulating cyclin B1and CDK1	[49]
	HO-8910, MCF-7, K562, and SMMC-7721 cells	Antiproliferation and cell cycle arrested at the G0/G1or the G2/M check points	Not referred	[48]
	HT-29 colon cancer cells and AGS gastric cancer cells	Antiproliferation and apoptosis induced	Increasing caspase-3 activity	[44]
Commodity provided by Korea Bio Polymer (KBP) company	AGS human gastric cancer cells.		Negatively regulating IGF-IR phosphorylation and inducing caspase-3 activation	[43]

**Table 2 molecules-24-04286-t002:** Anti-cancer activity and possible mechanisms of carrageenans.

Source	Target	Type of Activity	Possible Mechanisms	References
λ-carrageenan purchased from Sigma-Aldrich	B16-F10 and 4T1 bearing mice	Inhibition of tumor growth and improving immune system	Increasing the number of tumor-infiltrating M1 macrophages, DCs, and more activated CD4^+^ CD8^+^ T lymphocytes and enhancing the secretion of IL17A in spleen and significantly increase the level of TNF-α in tumor	[75]
Carrageenan oligosaccharides derived from *Kappaphycus striatum*	S180-bearing mice		Increase macrophage phagocytosis, the form of antibody secreted by spleen cells, spleen lymphocyte proliferation, NK cells activity, serumal IL-2 and TNF-a level	[76]
κ-carrageenan and λ-carrageenan purchased from Sigma-Aldrich	HeLa cells	Cell cycle delayed in G2/M phase or in both G1 and G2/M phase	Not referred	[56]
κ-selenocarrageenan consisted of selenium and κ-carrageenan	HepG2 cells	Cell cycle delayed in S phase	Upregulating Cyclin A and chk2 protein and down-regulating Cdc25A and cdk2 expression.	[77]
ι-Carrageenan	Human osteosarcoma cell line	Apoptosis induced and Cell cycle delayed in G1 phase	Downregulation of the Wnt/β-catenin signaling pathway through suppressing LRP6 expression and phosphorylation	[78]
κ-carrageenan oligosaccharides prepared from κ-carrageenan with enzyme	MCF-7 xenograft tumor	Antiproliferation and anti-angiogenic	Negative regulation of human VEGF, bFGF, bFGFR, and CD105	[79]

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
