# Peer review of "Anti-Cancer Activity of Porphyran and Carrageenan from Red Seaweeds"

_molecules, 2019, doi:10.3390/molecules24234286_

Round 1

Reviewer 1 Report

The paper describes the antitumoral activities of porphyran and carrageenan from red seaweeds.  Some suggestions to improve the paper are listed below:

Abstract. Line 22.: The indication "all in the family Rhodophyceae" is redundant. I suggest only :  all in Rhodophyceae.

References must be arranged according to the order of appearance in the text.

In the topic 2, the authors discussed the antitumor activity from red seaweeds, showing some examples of bioactive secondary metabolites such as phenolic compounds, halogenated monoterpenes, etc...  However, a powerful antitumoral compound, the halogenated monoterpene halomon was not mentioned.   This compound should be considered in the discussion of this specific topic. 

Line 141: Units of IC50 values should be checked.

Lines 188, 220 and 221: names of organisms should be written in italic form.

Figure 2. In the carrageenan structures, the number 3 in sulphate groups should be subscript.

In both tables the column Type of activity and possible mechanisms could be splited in two columns: Type of activity and Possible mechanisms. Thus, the organization of data could be improved.

An interesting review about oligosaccharides derived from red seaweed was published by Cheong et al., Molecules (2018), 23, 2451.  doi:10.3390/molecules23102451.   Why this reference was not cited in the paper ?

Reviewer 2 Report

The review entitled " Anti-cancer and anti-tumor activity of porphyran and carrageenan from red seaweeds" Zhiwei Liu1, 2 , Tianheng Gao3, ...and Xian Sun1, 5,  addresses a very interesting issue in the field of cancer, highlighting the potentiality of porphyran and carrageenan compounds obtained from red seaweeds as anti-tumor agents.

First the review gives an overview of the pharmacological properties of the main compounds such as polyphenols and sulphate polysaccharide extract from red seaweeds. Then, the authors analyzed in detail the anti-cancer activity of porphiran and carrageenenan and explored the possibility to use these molecule in cooperation with other anti-cancer chemotherapeutic agents.

It is a good review that manages to give a general information of the subject matter and the effects in different type of tumor.

Major comments: -

The authors have to use only "anti-cancer" or  "anti tumor" word in the title and during the review because they have the same meaning!

In order to improve the review we suggest to better discuss the examined compounds compared to those derived from brown seaweed. For example Fucoidan derived from the brown seaweed that also exhibits a significant anticancer activity against lung cell line  (Han et al.,  2008).

Minor comments:

line 316  we suggest to summarize the sentence........which had been damaged by drugs reducing side effect

line 217  We suggest to remove the last sentence line 317-319

line 131  Remove AND at the beginning of the line

 line 231 Remove Also

Reviewer 3 Report

In this manuscript, Liu et al reviews the potential anticancer interest of red seaweeds, particularly for compounds porphyrin and carrageenan. The subject is interesting, not only considering the high incidence of oncologic diseases but also to demonstrate the importance of marine products as a source of new bioactive compounds. In addition, as far as I could see, there is no similar published works. However, several problems can be detected which mus be solved before being acceptable for publication:

1) Title, abstract and text: why “anti-cancer and anti-tumor”? In parts of the text, it seems that these terms are completely different things. In other parts of the text, it seems that they are the same thing… In addition, as can be seen in lines 32-33, the definition of tumor can be confounded with the definition of cancer… Therefore, the authors must clarify which means each term and include this information in the text and, accordingly, perform the needed text changes in title, abstract, body text…

In this context, the sentence “” (line 112) is another evidence of the confusion between anticancer, antitumor, antiproliferative, cytotoxic,… terms.

2) In the abstract, the aim of the review should be better established – in my opinion, the last sentence of the introduction is not a complete reflex of the review

End of introduction: the focus of the review must be clarified as the authors are presenting not only mechanisms of action. In addition, the searching strategy, databases consulted, timespan covered in the search, keyword used in the search should be included

As the authors want to focus this work on mechanism of action, a figure including the main biochemical pathways affected by these compounds should be included in order to help the readers to better understand the given information

3) Writing mistakes – examples:

Anti-cancer vs anticancer

Anti-tumor vs antitumor

In vitro and In vivo should be written in italics

Other examples of written English problems: Lines 38 “the all”, 41, 43, 47-48, 54, 57, 129, 131, 313-314,…

Therefore all text must be carefully reviewed.

4) References – several problems can be detected:

-lines 30-33 – reference 1 is a 2013 article, however, in this part of the text, data from 2018 is presented!

-lines 35-36 – reference 2 is not adequate for the information given

Recent reviews which should be included:

https://www.mdpi.com/1420-3049/23/10/2451

https://www.sciencedirect.com/science/article/pii/S0144861717314066?via%3Dihub

Example of other important articles to be included:

https://www.mdpi.com/1660-3397/16/8/277

Authors are not following the order of numbering after first appearance in the text, for example: after reference 2 we can find reference 73? This must be reviewed thorough the text!

Round 2

Reviewer 3 Report

The authors improved the manuscript. However, several mistakes still can be detected - examples:

IC50 - the number must be underscript

line 140 - italics in R and S in the name of the compound

line 62

Therefore, all manuscript must be re-read and reviewed again to be improved.
